# Behind the Lines of #MeToo: Exploring Women's and Men's Intentions to Join the Movement

**Michela Menegatti \*** , **Silvia Mazzuca, Stefano Ciaffoni and Silvia Moscatelli**

Department of Psychology, University of Bologna, Viale Berti Pichat 5, 40126 Bologna, Italy
\* Correspondence: m.menegatti@unibo.it

**Abstract:** Campaigns supporting victims of gender harassment and abuse, such as #MeToo, have made and still make significant contributions to achieving the fifth UN Sustainable Development Goal aimed at eliminating "all forms of violence against all women". In two correlational studies, we examined possible antecedents of people's willingness to participate in the #MeToo campaign by focusing on the role of ambivalent beliefs toward women and men and the perceived effects of the movement. Men (Study 1) and women (Study 2) were asked to answer questions concerning hostile and benevolent beliefs about women and men, respectively, their perception of the beneficial and detrimental effects of #MeToo, and their intentions to participate in the campaign. Study 1 showed that men's hostile sexism toward women was associated with fewer intentions to actively support the #MeToo campaign and that the reduced beliefs that the movement had beneficial effects mediated this relation. Study 2 revealed that women's stronger benevolent beliefs about men were associated with decreased perception that the #MeToo campaign had a beneficial impact. In turn, such a perception was related to lower intentions to participate in supporting the campaign. Moreover, women's hostility toward men explained the intention to join the #MeToo movement through the mediation of the perception that the campaign was beneficial. The findings suggest that to foster participation in a feminist movement that promotes women's rights, it is necessary to eradicate traditional gender roles and the related ideologies that legitimate men's dominant position in society.

**Keywords:** ambivalent sexism; collective action; feminism; sexual harassment; gender equality; sustainable development goals

## 1. Introduction

The fifth UN 2030 Sustainable Development Goal aims to "achieve gender equality and empower all women and girls" through actions that, among others, "eliminate all forms of violence against all women and girls in public and private spheres" [1]. Estimates based on surveys from the period 2000–2018 and conducted across 161 countries and areas showed that "one in three women, nearly 736 million, have been subjected to physical violence, sexual violence or both by a husband or intimate partner, or sexual violence by a non-partner, at least once in their lifetime from 15 years of age". These data speak for themselves: violence against women is the most pervasive yet least visible human rights violation around the globe. Indeed, many sexual harassment and assault cases are not reported, mainly because the victims fear the possible consequences of filing a lawsuit or think they will not be believed [2]. More dramatically, because of such wide diffusion of sexual harassment, the phenomenon is normalized; thus, sometimes, victims are not even aware of having been violated [3]. Hence, several feminist movements arose to foster awareness of the widespread pervasiveness of sexual harassment and abuse against women, let victims know that they are not alone, and share the message that it is possible to take action against the perpetrators [4].

The most successful among these movements is, undoubtedly, #MeToo. As a matter of fact, the #MeToo campaign accomplished its goals. Victims of sexual abuse, harassment,

and mistreatment were encouraged to share their experiences and report their assaulters, and even laypeople who heard about the campaign reported being less dismissed of sexual assault [3,5–7]. Nevertheless, many people worldwide were critical of the movement, underlining its potential backlash for women's employability and career and even for relations between women and men [8,9]. In the face of such polarized attitudes, it is crucial to understand why some people are motivated to act to promote women's rights by supporting the #MeToo campaign, while others are not willing to participate in such actions. The present research addressed this issue by examining the role of sexist ideologies and favorable and unfavorable views of the #MeToo movement in accounting for individuals' intentions to engage in collective actions that support the campaign.

Although men generally show scarce participation in feminist collective movements—these movements might decrease their privileged status [10]—men's actions can be highly effective in obtaining a real social change [11]. Accordingly, Study 1 examined the role of sexism toward women and attitudes toward the #MeToo campaign in accounting for men's intentions to support the campaign.

Since women are far more likely than men to become victims of sexual violence and might directly benefit from feminist actions against sexual harassment and sexual abuse [10], Study 2 examined the correlates of intentions to support the #MeToo campaign among women. Specifically, it focused on women's ambivalence towards men, which is expressed through hostile resentment toward men's dominance on the one hand and respectful benevolence toward men who serve in the roles of providers and protectors on the other [12].

## 2. Ambivalence toward Women and Men as an Antecedent of Attitudes toward Sexual Harassment

Tolerance of sexual harassment and violence is nurtured by ideologies that acknowledge men's domination in society and emphasize the importance of women and men serving their traditional roles in society. Among these ideologies, ambivalent sexism toward women seems to be the most important predictor of people's attitudes concerning the sexual abuse of women. According to Glick and Fiske [13], ambivalent sexism toward women represents the "stick and carrot" that contribute to maintaining women's subordinate role in society. The "stick" is represented by hostile sexism, an adversarial view of women who refuse to embrace traditional gender roles and are considered as seeking to gain power over men. Benevolent sexism, the "carrot," constitutes a paternalistic view of women who conform to traditional gender roles as pure creatures in need of men's protection. This portrayal of women seems positive but implies that women are weak and suited for subordinate gender roles.

Several studies have found that ambivalent sexism predicts tolerance of sexual harassment and rape myth acceptance, that is, the tendency to blame the victims of sexual harassment and rape and absolve the perpetrator [14–17]. Interestingly, hostile sexism toward women is also associated with men's higher proclivity to engage in sexual aggression; this association is explained by the belief that women victims actually wanted to take part in the sexual encounter but pretended not to [18].

Importantly for the present contribution, Moscatelli et al. [19] found that higher hostile sexism toward women predicted women's higher skepticism toward the allegations made by the victims of the so-called "Weinstein scandal" and more negative views of the #MeToo campaign. Similarly, hostile sexism was associated with more negative beliefs concerning the outcomes of the movement, which in turn were associated with a higher tendency to consider social-sexual behaviors as sexual harassment [6].

Research concerning the role of benevolent sexism toward women in predicting attitudes toward sexual harassment provided less consistent findings. On the one hand, some studies found that benevolent sexism was not a significant predictor of rape myths acceptance and even predicted support for gender equality [20,21]. On the other, further findings showed that people holding benevolent sexism toward women were more likely

to blame the victims of a rape perpetrated by an acquaintance who is a man [18] while reporting lower rapist blaming and shorter recommended prison sentences for rape perpetrators [22]. A recent meta-analysis [23] has found that both hostile and benevolent sexism toward women independently and significantly affects attitudes that justify and legitimize violence against women. However, if hostile sexism's contribution is substantial, benevolent sexism's contribution is relatively small.

Besides sexist ideologies regarding women, attitudes towards sexual harassment could be affected by ambivalent beliefs about men. According to Glick and Fiske [12], women respond to men's power and dominance by holding both hostile and benevolent beliefs about men. Hostility conveys resentment toward men's higher status and sexual aggressiveness. In contrast, benevolent beliefs about men represent admiration for their role as protectors and providers by expressing the maternalistic view that men need women's care [24]. It has been noted, however, that even ambivalence toward men can be a means through which men's domination over women within a society is justified and maintained. Indeed, hostile beliefs about men might represent a "safe jab" that allows expressing discontent on how men treat women while acknowledging their inevitable higher status. At the same time, the benevolent portrayal of men as needing maternalistic care implicitly recognizes men's superiority in less private (and more socially acknowledged) domains [12,14].

To date, little evidence has been collected on the relation between hostile and benevolent beliefs about men and attitudes towards sexual harassment and violence. Sakalli-Uğurlu et al. [25] found that people with higher ambivalence toward men tend to consider sexual harassment a less severe problem. In addition, Chapleau et al. [26] pointed out that benevolent beliefs—but not hostile beliefs—about men were related to higher rape myth acceptance, especially in women respondents. Similarly, results collected in an Italian sample showed that people who endorse benevolent beliefs about men also think that the perpetrator did not actually intend to rape ("He didn't mean to") [24] and do not consider emotional abuse as violence [26].

### 3. The #MeToo Movement

The public discourse on sexual harassment has dramatically increased since the explosion of the social media #MeToo movement [27]. The movement spread worldwide in 2017 when journalists reported decades of sexual harassment and assault by Hollywood producer Harvey Weinstein. Some days after, actress, producer, and activist Alyssa Milano asked those who had experienced sexual harassment to reply with "me too" to her tweet or post #MeToo as social media status. Within 24 h, the hashtag #MeToo became a global collective movement and revealed the prevalence of sexual harassment and sexual assault experiences for famous as well as ordinary women (and men) in various contexts [28]. It also generated subsequent social media movements, such as #TimesUp (anti-gender discrimination movement), #BelieveWomen (solidarity with survivors of sexual assault), #UsToo (anti-harassment of minority groups in the workplace), and #HimToo (platform for men victims of sexual harassment or assault and men falsely accused of sexual misconduct). In 2020 Weinstein was sentenced to 23 years in prison for rape and sexual assault, and he is currently facing a second trial in another state.

Men, on their part, have started campaigns, such as #IHave, #IDidThat, and #IWill, through which they admitted to having witnessed, condoned, or engaged in sexual harassment and committed to changing their views on the matter. As a consequence of #MeToo, global initiatives (for example, HeForShe, Men Advocating Real Change, Token Man, and The Good Lad Initiative) have recently called on men to take actions that challenge gender inequality in solidarity with women.

The positive effects of the #MeToo campaign are unquestionable. A large body of research has shown that it awakened a greater public awareness of gender and power dynamics in sexual harassment and provided victims with new ways of support [29]. It also led to sexual abuse and assault disclosures in unprecedented ways. For instance, Alaggia and Wang [3] found that victims reported feelings of freedom raised by speaking up their

truth and that many people could recognize that they had been victims of sexual abuse or assault only after reading or watching news concerning #MeToo.

The #MeToo movement also affected the extent to which people considered social-sexual behaviors in the workplace, such as sexist jokes, repeated invitations to date following a polite rejection, or hugs lasting "a little too long" as sexual harassment [6,30]. Through a four-wave longitudinal study, Szekeresa et al. [7] demonstrated that lay people's dismissal of sexual assault decreased following the campaign. Interestingly, working men interviewed by Atwater et al. [8] reported that they would be more careful about potentially inappropriate behavior and less likely to engage in sexual harassment in the future.

Despite its merits, the #MeToo campaign has also been extensively criticized for its possible detrimental effects. Some people believe that the movement went too far and ruined the lives of men who were publicly accused of sexual misconduct without a fair process, increased false complaints, and fed calumny and androphobia [9,31]. It has also been accused of failing to distinguish between different degrees of sexual misconduct and eroding the relations between men and women [32].

These critical views of the movement can harm women. For instance, in the Atwater et al.'s [8] study mentioned above, fifteen percent of respondents indicated that they would be reluctant to hire women especially attractive women for jobs that required close men–women interactions. These data are in line with recent findings showing that people with negative attitudes about the outcomes of #MeToo were less likely to perceive subtle forms of sexual harassment (e.g., jokes, excessive attention) as such [6].

In the attempt to understand the motivations underlying criticism of or support for the #MeToo campaign, research has looked into people's beliefs about gender roles. For instance, Kende et al. [33] found that lower levels of gender system justification the tendency to legitimize gender inequality in a given society [34] were associated with a greater perception of the campaign as empowering for women. Moreover, men endorsing gender system justification reported that the #MeToo campaign wrongfully stained men's reputations. Similarly, Kunst et al. [31] found that men's tendency to score higher than women on hostile sexism toward women and rape myths acceptance explained why they were less favorable toward #MeToo than women.

Overall, the reported findings clearly showed that individuals' attitudes towards sexual harassment and the #MeToo campaign are related to the endorsement of sexist ideologies. However, to our knowledge, no research has analyzed whether ambivalence about both women and men is associated with active participation in the movement.

## 4. Research Aims

The present research aimed to investigate the relationship between ambivalent beliefs about women (Study 1) and men (Study 2) and intentions to engage in collective action supporting the #MeToo campaign among men (Study 1) and women (Study 2). Moreover, we tested whether these associations were mediated by the perception that the movement was beneficial for raising awareness of sexual harassment and giving voice to the victims or, conversely, the belief that it was detrimental to men and spoiled the harmonious relations between sexes.

To this aim, we conducted two studies in Italy, one of Europe's most masculine and gender-unequal countries, with extensive sexist attitudes and widespread patriarchal views of sexual abuse [35]. Recent national reports [36] indicated that the most pervasive gender stereotypes among Italians are "Being successful at work is more important for men than for women," "Men are less suited for housework", and that these stereotypes are shared by the 58.8% of the population. For what concerns sexual violence, 39.3% of Italians believe that a woman could avoid sexual intercourse if she really wanted to, 23.9% declared that women could provoke sexual violence by their clothing, and 15.1% think that a woman who suffered sexual abuse when she was drunk or on drugs is at least in part responsible. These stereotypes are evident in Italian media, whereby women are more often portrayed as objectified, attractive, and dressed in a sexualized way than in less masculine and gender-

unequal countries [35]. Unfortunately, these stereotypic representations have worsened since the nineties [37]. It is also important to consider that Italy still represents, to a certain extent, a "culture of honor" that valorizes norms of superiority and toughness for men and norms of modesty and shame for women [38,39]. According to these norms, women should avoid behaviors that might threaten a family's reputation, and men's violence against women might be even justified when aimed at preserving the integrity of the man and the family [39,40].

In this context, it is not surprising that the public debate tried to depict the #MeToo movement and its local counterpart #QuellaVoltaChe (literally, #ThatTimeWhen) as a campaign against powerful men rather than a movement in support of all victims of sexual abuse [41,42]. Consequently, the reactions to the campaign were tepid or even hostile, #MeToo "has been flooded under an impressive wave of miscredit, discredit, and misogyny" [42], and the few public figures who explicitly supported it (such as the actress Asia Argento) have been harshly attacked [43,44]. In similar contexts, it is interesting to understand why some people concretely supported the #MeToo campaign while others seemed indifferent or even hostile.

## 5. Study 1

Men's involvement in feminist movements is rather scarce, as they are more inclined to see such movements as harmful [33]. Yet, their support is essential to obtaining real change. Indeed, men are less likely than women to be seen as self-interested, and their higher societal position could make their actions more effective [11,45]. For these reasons, Study 1 examined the antecedents of men's intentions to endorse collective action in favor of the #MeToo campaign. We investigated whether men's hostile and benevolent sexism toward women would predict their intentions to participate in #MeToo and whether this relation would be accounted for by their perceptions of the beneficial and detrimental effects of the campaign.

As mentioned, previous research highlighted that endorsing hostile—but not benevolent sexist beliefs toward women was associated with lower perceived benefit and higher perceived harm of the #MeToo campaign [30,31]. Accordingly, we predicted that men's levels of hostile sexism toward women would be negatively associated with the perception that the movement was beneficial for challenging sexual harassment (Hypothesis 1a) and positively associated with the perception that the #MeToo campaign was detrimental (Hypothesis 1b). Given that people are more willing to engage in collective action to support a disadvantaged group when they believe that such action will help them achieve the group's goals [46], we predicted that perceiving the #MeToo movement as beneficial to reducing sexual harassment and abuse would be positively associated with the intentions to participate in the movement (Hypothesis 2a). In the same line of reasoning, believing that the #MeToo campaign had detrimental effects should be negatively related to the intention to support the campaign (Hypothesis 2b).

Previous work found that men who endorsed hostile sexist beliefs against women were less willing to engage in collective action in favor of gender equality [47]. More generally, men are more prone to support actions that protect women from violence rather than participate in feminist collective movements [48]. Thus, we expected that men with higher levels of hostile sexism toward women would be less willing to participate in collective action in favor of the #MeToo campaign (Hypothesis 3). This association should be mediated by their belief that #MeToo had beneficial (Hypothesis 4a) and detrimental effects (Hypothesis 4b).

Previous evidence is mixed regarding benevolent sexism toward women. Some studies showed associations with acceptance of sexual harassment and rape myths [18,22], whereas others were unable to observe any relation [20]. For these reasons, we did not advance specific hypotheses concerning the effects of benevolent sexism.

*5.1. Method*

5.1.1. Participants

Both Study 1 and Study 2 were approved by the Bioethical committee of the authors' university. Data from both studies were collected in September–December 2021 using the Qualtrics platform.

Participants of Study 1 were recruited among students enrolled in various courses of a large University of Northern Italy and through social media and snowball sampling. Of 390 participants who filled in the questionnaire, 14 were excluded from the analyses as they did not fill in all the study variables or they did not indicate their sex. The final sample consisted of 374 men, all white Italian, ranging from 18 to 60 years old (M = 22.41, SD = 4.80). Based on Kelloway's [49] recommended 5:1 ratio of observation to estimated parameters for SEM with latent variables, the sample size is adequate as the model has 57 parameters.

5.1.2. Procedure and Measures

Participants were assured of the anonymity of the data and provided informed consent before filling in the questionnaire. They were then presented with measures of hostile and benevolent sexism towards women (in randomized order), beliefs concerning the beneficial and detrimental effects of the #MeToo campaign (in randomized order), and intentions to support and participate in the movement. In the end, participants were asked their gender (confirming that all respondents were men), age, and sexual orientation. Completing the questionnaire required approximately 10 min.

To assess hostile and benevolent sexism toward women, we used six items from the Italian short version of the Ambivalent Sexism Inventory (ASI) [50]. The items were selected as their factor loadings were > 0.50 (see Kosakowska-Berezecka et al., for a similar procedure) [51]. Three items measured benevolent sexism towards women and assessed positive but patronizing attitudes toward women (e.g., "Women should be cherished and protected by men"; $\alpha$ = 0.53). Given that other studies have found relatively low internal reliability consistency for benevolent sexism toward women (and hostile beliefs about men) when measured with a 3-item scale [52], and that alphas from 0.5 to 0.7 could be considered acceptable for 3-item scales [53,54], we decided to keep benevolent sexism as a variable in the model. The other three items measured hostile sexism toward women in terms of adversarial views of women (e.g., "When women lose to men in a fair competition, they typically complain about being discriminated against"; $\alpha$ = 0.73). Responses were given on scales ranging from 0 (strongly disagree) to 5 (strongly agree). Higher scores indicated stronger benevolent or hostile sexism towards women.

Then the questions concerning the #MeToo movement were introduced by asking participants if they knew what #MeToo was. All participants confirmed that they knew the campaign. The perception that the #MeToo campaign had beneficial effects was measured by asking participants to indicate the extent to which they agreed (1 = strongly disagree; 7 = strongly agree) with three items concerning the importance of the movement for the victims of sexual harassment and society at large. Two items were drawn from Kunst et al. [31] ("The campaign is important because it gives victims of sexual assault a voice;" "The campaign is important because it makes it easier for victims of sexual assault to out themselves") and one item was developed ad-hoc ("Being part of the #MeToo movement can be useful for a social change"). Overall, the scale had good reliability, $\alpha$ = 0.82.

The perception that the #MeToo campaign had detrimental effects was assessed through three items drawn from Kunst et al. [31], which concerned the extent to which participants agreed (1 = strongly disagree; 7 = strongly agree) with the idea that the campaign fueled a hostile climate between women and men and exaggerated sexual harassment accusations: "The campaign legitimates false accusations"; "The movement wrongfully labels a lot of people as sexual assaulters"; The movement creates an exaggerated vigilantism/witch hunt" ($\alpha$ = 0.85).

Finally, participants' intentions to participate in the #MeToo movement were measured through four items. Participants were asked to indicate the extent to which they were willing to take part in a series of actions (1 = not willing at all; 7 = strongly willing): "Participate in events related to the #MeToo movement (marches, demonstrations, etc.)"; "Participate in a flash mob to support the #MeToo campaign"; "Participate in a rally in favor of #MeToo"; "Write something about #MeToo on Facebook" ($\alpha$ = 0.92).

See Supplementary Material for Exploratory Factor Analysis (EFA) conducted to determine the factor structure of measures concerning the #MeToo campaign.

### 5.1.3. Data Analysis

After conducting preliminary analyses that confirmed that the normality assumptions were not violated, we performed exploratory factor analysis (see Supplementary Material). Then, we performed correlations to assess the relationships between measures using the Statistical Package for the Social Sciences (SPSS 27.0). To test the hypotheses, we estimated a model in which the two measures of sexism against women were included as predictors, the perception that the #MeToo campaign was beneficial and detrimental as mediators, and collective action intentions as the outcome variable. To adjust for measurement error, we conducted SEM with latent factors on the software Mplus 8.3 [55], using items as indicators loading onto the several variables of interest to this study. Model parameters were estimated using the maximum likelihood (ML) method. To test for mediation, bootstrap (5000 resamples) estimates of indirect effects and bootstrapping bias-corrected confidence intervals (CIs) were calculated. The model fit was evaluated using multiple indices: the Comparative Fit Index (CFI) and the Tucker–Lewis Index (TLI), with values higher than 0.90 indicative of an acceptable fit and values higher than 0.95 indicative of an excellent fit; and the Root Mean Square Error of Approximation (RMSEA), with values below 0.08 indicative of an acceptable fit and values less than 0.05 representing a very good fit [56]. In addition, we inspected the 90% confidence interval of the RMSEA: when the upper bound of this confidence interval is ≤0.10, the model fit can be considered acceptable.

### 5.2. Results

Descriptive statistics and correlations among study variables are shown in Table 1.

**Table 1.** Descriptive statistics and correlations among Study 1 variables.

| Variables | Descriptive Statistics | Correlations | | | | |
| | M (SD) | 1. | 2. | 3. | 4. | 5. |
| --- | --- | --- | --- | --- | --- | --- |
| 1. Benevolent Sexism toward Women | 2.26 (1.11) | – | 0.315 *** | −0.065 | 0.013 | −0.080 |
| 2. Hostile Sexism toward Women | 1.82 (1.07) | | – | −0.292 *** | 0.342 *** | −0.292 *** |
| 3. Beneficial Effects #MeToo | 4.99 (1.07) | | | – | −0.485 *** | 0.483 *** |
| 4. Detrimental Effects #MeToo | 3.69 (1.25) | | | | – | −0.305 *** |
| 5. Participation #MeToo | 3.22 (1.53) | | | | | – |

Note. *** $p < 0.001$.

The results of the SEM analysis indicated that the model tested fitted the data well, CFI = 0.938, TLI = 0.921, RMSEA = 0.066, CI [0.056, 0.076]. Table 2 reports the estimates for direct and indirect effects and the CIs of the main model.

Hostile sexism toward women was negatively associated with the perception that the #MeToo campaign had beneficial effects (Hypothesis 1a) and positively associated with the belief that the movement had a detrimental impact (Hypothesis 1b). The association between the perception that the #MeToo campaign had beneficial effects and intentions to support the movement was positive and significant (Hypothesis 2a). In contrast, the association between the perception that the movement had detrimental effects and the outcome was not significant (Hypothesis 2b). Supporting Hypothesis 3, hostile sexism had

a significant, negative direct link with intentions to participate in the #MeToo campaign. The relation between hostile sexism toward women and lower intentions to act in favor of the #MeToo campaign was mediated by the decreased perception that the movement had beneficial effects, as shown by the significant indirect effect (Hypothesis 4a). In contrast, there was no significant mediation by the perception that the #MeToo campaign had a detrimental impact (Hypothesis 4b).

**Table 2.** Standardized direct and indirect effects of the SEM of Study 1.

| | Beneficial Effects #MeToo | Detrimental Effects #MeToo | Participation #MeToo |
|---|---|---|---|
| | $\beta$ (SE) [95% CI] | $\beta$ (SE) [95% CI] | $\beta$ (SE) [95% CI] |
| Direct effects | | | |
| Benevolent Sexism toward Women | 0.106 (0.109) [−0.088, 335] | −0.277 ** (0.106) [−0.496, −0.077] | 0.030 (0.096) [−0.150, 0.224] |
| Hostile Sexism toward Women | −0.435 *** (0.100) [−0.644, −0.249] | 0.566 *** (0.104) [0.367, 0.778] | −0.021 * (0.103) [−0.427, −0.011] |
| Beneficial Effects #MeToo | | | 0.434 *** (0.085) [0.278, 0.611] |
| Detrimental Effects #MeToo | | | 0.002 (0.090) [−0.164, 0.189] |
| Indirect effects | | | |
| Benevolent Sexism toward Women → Beneficial effects #MeToo | | | 0.046 (0.051) [−0.037, 0.166] |
| Benevolent Sexism toward Women → Detrimental Effects #MeToo | | | −0.001 (0.028) [−0.063, 0.051] |
| Hostile Sexism toward Women → Beneficial effects #MeToo | | | −0.189 ** (0.063) [−0.334, −0.092] |
| Hostile Sexism toward Women → Detrimental Effects #MeToo | | | −0.001 (0.054) [−0.094, 0.118] |

Note. * $p < 0.05$. ** $p < 0.01$. *** $p < 0.001$.

Regarding benevolent sexism toward women, the results showed no significant link with the perception that the #MeToo campaign had beneficial effects and a positive association with the perception that the movement was detrimental for women and men. The relation between benevolent sexism and intention to participate in the movement was not mediated by either the perception that #MeToo had beneficial or detrimental effects. Figure 1 displays the model and direct effects.

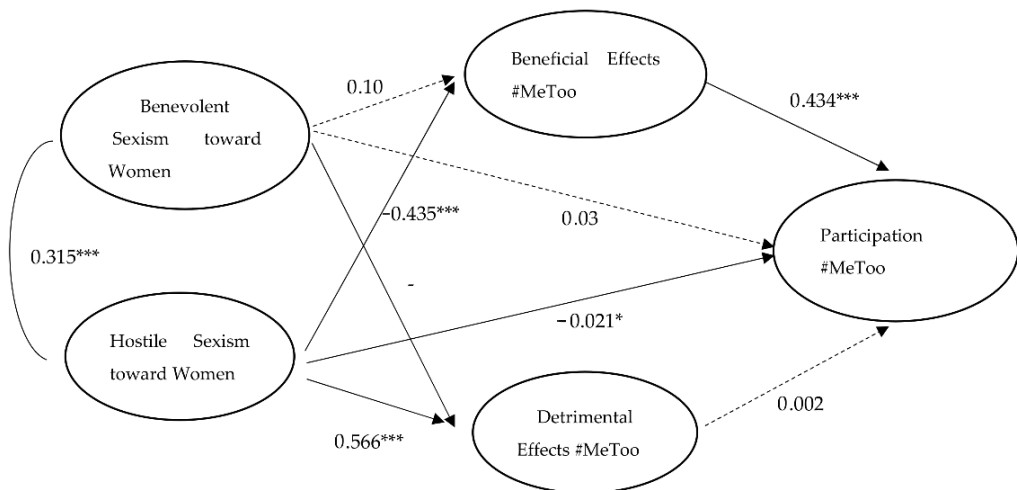

**Figure 1.** Standardized solution of the model testing the relations among men's benevolent and hostile sexism toward women, beneficial and detrimental effects of the #MeToo campaign, and intentions to participate in the campaign (Study 1). * $p < 0.05$. *** $p < 0.001$.

## 6. Study 2

Study 2 was conducted among women, who are more often victims of sexual harassment and could therefore feel closer to the cause around which the #MeToo campaign revolves. However, despite feminist movements drawing attention to gender inequity, women still face barriers to participating in these forms of collective action that come from their group membership [48]. In Study 2, we analyzed some of these barriers focusing on women's ambivalence toward men a relatively neglected issue and their attitudes towards the #MeToo campaign. Specifically, we examined whether and how women's hostile and benevolent beliefs toward men were associated with their intentions to engage in collective actions to support the #MeToo movement. We also analyzed whether this relationship was mediated by women's perception that the #MeToo campaign had beneficial or harmful effects on women, men, and society.

Evidence shows that benevolence toward men was associated with less negative attitudes toward sexual harassment and abuse [14] and a stronger tendency to excuse the perpetrator [24]. Coherently, we expected that women's levels of benevolent beliefs about men would be negatively associated with the perception that the movement had beneficial effects (Hypothesis 5a) and positively associated with the perception that #MeToo was detrimental (Hypothesis 5b). As in Study 1, we also expected that believing that the #MeToo campaign was beneficial to reducing sexual harassment and abuse would be positively associated with the intentions to engage in actions that support the movement (Hypothesis 6a). In contrast, believing that the movement had detrimental effects should be negatively related to the intention to participate in the campaign (Hypothesis 6b).

A fundamental antecedent of collective action on behalf of a group is perceiving that the group is mistreated [46]. In the specific case of feminist movements, sexist ideologies are crucial to this perception as they legitimize the status quo and have a sedative effect on people's willingness to advocate social change [48]. Consequently, we predicted that women endorsing more benevolent views of men—which justify men's higher status as preordained would be associated with a lower intention to engage in collective action that supports the #MeToo movement (Hypothesis 7). Since people participate in collective movements especially when they perceive them as effective in advocating the rights of the disadvantaged group [46], we also predicted that the perception that the #MeToo campaign was beneficial and detrimental would mediate the association between women's benevolence toward men and their willingness to participate in the campaign (Hypothesis 8a,b). Given the mixed evidence on the effects of hostile beliefs about men on people's attitudes toward sexual harassment [14,25,26], we did not advance specific hypotheses concerning women's hostility toward men.

### 6.1. Method

#### 6.1.1. Participants

Three-hundred-eleven participants, all white Italian, were recruited among students at the same University as Study 1 and through snowball sampling. Among them, 17 did not indicate their gender or did not fill in all the study variables and were therefore removed from the analyses. The final sample consisted of 294 women, all white Italian, ranging from 18 to 59 years old ($M$ = 22.07, $SD$ = 5.56).

#### 6.1.2. Procedure and Measures

After being assured of the anonymity of the data and providing informed consent, participants filled in a questionnaire containing the same measures of Study 1 except for hostile and benevolent sexism toward women, which was replaced by hostile and benevolent beliefs about men. Ambivalence toward men was measured using six items from the Italian short version of the Ambivalence Toward Men Inventory (AMI) [47]. The items were selected as their factor loadings were >0.50 as in Study 1.

Three items measured hostile beliefs in terms of resentment toward men's dominance over women (e.g., "Most men sexually harass women, even if only in subtle ways, once

they are in a position of power over them"; α = 0.52). As mentioned in Study 1, the relatively low reliability is consistent with other studies which measured hostile beliefs about men with a 3-item scale [52] and alphas from 0.5 to 0.7 could be considered acceptable for 3-item scales [50,51]. The other three items measured benevolent beliefs, which capture respondents' appreciation of men as providers and protectors (e.g., "Every woman needs a male partner who will cherish her"; α = 0.62). Responses were given on scales ranging from 0 (strongly disagree) to 5 (strongly agree). Higher scores indicated stronger benevolent and hostile beliefs about men. The perception that the #MeToo campaign had beneficial (α = 0.83) and detrimental effects (α = 0.75), as well as intentions to participate in the movement (α = 0.89), were assessed as in Study 1. See Supplementary Material for Confirmatory Factor Analysis (CFA) conducted to confirm the factor structure of measures concerning the #MeToo campaign.

### 6.1.3. Data Analysis

We first conducted preliminary analyses that confirmed that the data's normality assumptions were not violated using SPSS 27.0. We then performed confirmatory factor analysis (see Supplementary Material) and correlations among variables. The hypotheses were tested using Mplus 8.3. We estimated a model in which hostile and benevolent beliefs about men were included as predictors, the perception that the #MeToo campaign was beneficial and detrimental as mediators, and intentions to participate in the campaign as the outcome variable. As in Study 1, we conducted SEM with latent factors and estimated model parameters using the ML method. Bootstrap (5000 resamples) estimates of indirect effects and bootstrapping bias-corrected confidence intervals (CIs) were calculated to test for mediation. The model fit was evaluated using CFI, TLI, and RMSEA as specified in Study 1.

### 6.2. Results

Descriptive statistics and correlations among study variables are displayed in Table 3.

**Table 3.** Descriptive statistics and correlations among Study 2 variables.

| Variables | Descriptive Statistics | Correlations | | | |
| --- | --- | --- | --- | --- | --- |
| | M (SD) | 2. | 3. | 4. | 5. |
| 1. Benevolent Beliefs about Men | 0.87 (0.81) | 0.130 * | −0.277 *** | 0.419 *** | −0.224 ** |
| 2. Hostile Beliefs about Men | 3.00 (0.92) | – | 0.091 | 0.009 | 0.033 |
| 3. Beneficial Effects #MeToo | 5.58 (1.03) | | – | −0.508 *** | 0.509 *** |
| 4. Detrimental Effects #MeToo | 2.62 (1.23) | | | – | −0.314 *** |
| 5. Participation #MeToo | 4.33 (1.61) | | | | – |

Note. * $p < 0.05$. ** $p < 0.01$. *** $p < 0.001$.

The fit of the estimated model was very good, CFI = 0.949, TLI = 0.936, RMSEA = 0.067, CI [0.055, 0.078]. Table 4 reports the estimates for direct and indirect effects and the CIs of the main model.

As hypothesized, women's benevolent beliefs about men were significantly related to the proposed mediators: The association with the perception that the #MeToo campaign had beneficial effects was negative (Hypothesis 5a), whereas the relation with the perception that the campaign had detrimental effects was positive (Hypothesis 5b). Supporting Hypothesis 6a, there was a significant positive association between the perception that the #MeToo campaign had beneficial effects and the intention to endorse collective actions that support the movement. However, the association between the perception that the #MeToo campaign had detrimental effects and the outcome was not significant (Hypothesis 6b). The direct link between benevolence toward men and intentions to participate in the campaign was not significant (Hypothesis 7). As expected, benevolent beliefs about men were related to lower intentions to support the #MeToo campaign through the mediation of reduced perception that the movement had beneficial effects, as revealed by the significant

indirect effect (Hypothesis 8a). Perceiving that the #MeToo campaign was detrimental did not mediate the relation between benevolence toward men and intentions to join it (Hypothesis 8b).

**Table 4.** Standardized direct and indirect effects of the SEM for Study 2.

| | Beneficial Effects #MeToo | Detrimental Effects #MeToo | Participation #MeToo |
|---|---|---|---|
| | *ß* (SE) [95% CI] | *ß* (SE) [95% CI] | *ß* (SE) [95% CI] |
| **Direct effects** | | | |
| Benevolent Beliefs about Men | −0.387 *** (0.098) [−0.567, −0.181] | 0.564 *** (0.082) [0.395, 0.712] | −0.101 (0.085) [−0.275, 0.062] |
| Hostile Beliefs about Men | 0.185 ** (0.065) [0.050, 0.307] | −0.125 * (0.037) [−0.238, −0.008] | −0.071 (0.056) [−0.182, 0.040] |
| Beneficial Effects #MeToo | | | 0.529 *** (0.089) [0.351, 0.702] |
| Detrimental Effects #MeToo | | | 0.024 (0.100) [−0.172, 0.217] |
| **Indirect effects** | | | |
| Benevolent Beliefs about Men → Beneficial Effects #MeToo | | | −0.204 ** (0.070) [−0.355, −0.081] |
| Benevolent Beliefs about Men → Detrimental Effects #MeToo | | | 0.014 (0.058) [−0.101, 0.130] |
| Hostile Beliefs about Men → Beneficial Effects #MeToo | | | 0.198 * (0.040) [0.025, 0.182] |
| Hostile Beliefs about Men → Detrimental Effects #MeToo | | | −0.003 (0.014) [−0.034, 0.024] |

Note. * *p* < 0.05. ** *p* < 0.01. *** *p* < 0.001.

Hostile beliefs about men were positively associated with the perception that the #MeToo campaign was beneficial and negatively associated with the belief that the movement had detrimental effects. The direct link between hostility toward men and the intentions to participate in the campaign was not significant. Hostility toward men was positively associated with intentions to support the campaign, with increased perception that the movement had beneficial effects mediating this association. The perception that the #MeToo campaign was detrimental did not mediate the association between hostile beliefs and intentions to participate. Figure 2 displays the model and the direct effects.

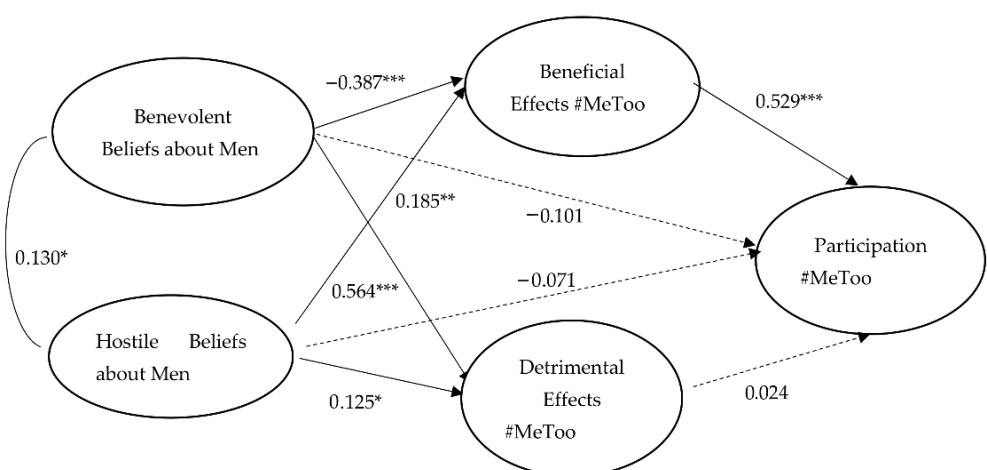

**Figure 2.** Standardized solution of the model testing the relations among women's benevolent and hostile beliefs about men, beneficial and detrimental effects of the #MeToo campaign, and intentions to participate in the campaign (Study 2). * *p* < 0.05. ** *p* < 0.01. *** *p* < 0.001.

## 7. General Discussion

The present studies aimed to investigate possible antecedents of men's and women's willingness to engage in collective action that supports the #MeToo movement by focusing on the role of ambivalent beliefs about women and men. Moreover, we tested whether the relation between ambivalence and intentions to participate in the campaign was mediated by the perception that #MeToo had beneficial and detrimental effects.

Study 1 supported our predictions that men's hostile sexism toward women was associated with lower intentions to act in favor of the #MeToo movement and that this relation was mediated by the reduced perceived beneficial effects of the campaign. Although higher hostility toward women was related to men participants' stronger perception that the movement was detrimental to men and society at large, such a perception did not account for the relation between hostile sexism and participants' active support of the campaign. The role of men's benevolent sexism toward women is less clear-cut. Men who believed that women are pure creatures who need their protection perceived the #MeToo campaign as more damaging, probably because it disrupts the traditional relations between sexes. However, this perception was not related to their willingness to engage in collective action supporting the campaign.

Study 2 showed that women's benevolent beliefs toward men, according to which men should be admired and women should take care of them in the domestic realm in compensation for their protection, were associated with a lower perception of the beneficial effects of the #MeToo campaign, which in turn was related to lower intentions to participate in the movement. Moreover, hostility toward men was associated with intentions to participate in the campaign through the mediation of its perceived beneficial effects. In other words, women who felt more resentment toward men's dominant position in society and believed that men in positions of power often sexually harassed women were also more likely to recognize the beneficial effects of the campaign and, therefore, to act to support it. In line with Study 1, although benevolent and hostile beliefs about men were significantly related to the perception of the movement as damaging, such a perception did not affect women's intentions to participate in the campaign.

### 7.1. Theoretical and Practical Implications

The present research provided novel evidence on why some people are more willing than others to support feminist collective actions. Indeed, it showed that intentions to participate in the #MeToo campaign strongly depend on men's and women's endorsement of sexist ideologies that embrace a patriarchal view of society and justify men's domination over women. It also highlighted that the perceived beneficial effects of the movement explained the relation between sexist ideology and intentions to support the movement collectively.

About men, this research brings novel evidence that the more they tend to share hostile attitudes toward women who defy traditional gender roles, the less they are inclined to participate in the #MeToo movement because they see it as less beneficial for the victims of sexual harassment and for changing the status quo. These results add to previous evidence that hostile sexism toward women is a crucial predictor of tolerance toward sexual harassment [15–17], rape myth acceptance [14], and attitudes toward the #MeToo campaign [19,20]. Indeed, we found that hostile sexism affected not only men's attitudes toward #MeToo but also their intentions to participate in the movement actively.

Besides, it is worth noting that men who were more inclined to think that women seek to control men—through sexuality or feminist ideology—tended to consider the #MeToo movement damaging to men and the relations between sexes. This finding is consistent with the public debate that tried to depict the campaign as a "witch-hunt" pursued by feminists against powerful men or a way for women to gain popularity. However, surprisingly, this view of #MeToo did not predict participants' intentions to support the movement. This finding may suggest that, for our participants, minimizing the #MeToo campaign's effectiveness in supporting the victims was more acceptable than expressing negative

attitudes toward the campaign itself, for example, stating that it created exaggerated vigilantism or that it legitimized false accusations.

The results also revealed that men who considered women pure and sensible creatures that need their protection felt that the #MeToo campaign was more damaging—but not more beneficial, probably because the movement threatened their view of women as refined ladies who are not supposed to behave in a way that makes themselves vulnerable to sexual harassment. This is clearly in line with previous evidence that higher benevolent sexism was associated with more negative attitudes toward sexual harassment [18] and the #MeToo campaign [31]. However, neither men's benevolent sexism toward women nor negative attitudes towards the #MeToo campaign was related to their intentions to act in favor of the movement, a finding that is partially consistent with previous studies according to which benevolent sexism has a little impact on attitudes that justify violence against women [23]. These findings could also be explained considering that most men perceive #MeToo as a movement that concerns only women who failed to meet gender expectations, such as those working in show business or feminists. Given that men endorsing sexist views of women often share both hostile and benevolent attitudes at once [13], the protection from violence granted by benevolent sexism ceases to exist for women who support or are involved in the #MeToo movement. In such a case, hostile sexism comes into play and becomes more critical in predicting men's attitudes and behaviors toward the movement [23].

Additionally, our findings provide novel and fertile ground in the literature on feminist collective action and the #MeToo campaign by considering the critical role of ambivalent beliefs about men as a factor that partially explains women's intentions to participate in a feminist movement. Indeed, previous research has been focused almost exclusively on ambivalent sexism toward women and its association with attitudes concerning sexual harassment in general [14–17,26] and #MeToo in particular [31]. Only a few studies have investigated the relationship between benevolent beliefs about men and attitudes toward sexual harassment and abuse [14,24]. We uniquely revealed that women's ambivalence toward men might increase their participation in feminist movements that supports the victims of sexual harassment and highlighted that this effect is explained by the perceived effectiveness of the #MeToo campaign to achieve this aim.

Interestingly, women's hostile and benevolent beliefs about men were the opposite in explaining their attitudes toward the #MeToo campaign and, in turn, their intentions to collectively support the movement. The more women were aware of and resented men's dominant position in society, the more likely they were to endorse collective action that challenged the current situation through perceiving #MeToo as more helpful. This effect is particularly interesting considering that hostile beliefs about men also imply the idea that, when in a position of power, men sexually harass women. Indeed, the #MeToo campaign had its worldwide rise precisely when several women accused the film producer Harvey Weinstein of sexual harassment, assault, and rape. Reasonably, it is not surprising that endorsing hostile beliefs about men increased the perceived benefit of #MeToo and the active support for the movement. Conversely, women who admire men for their higher status and believe that men need women's care seem to fail to perceive the positive outcomes of the #MeToo campaign, probably because they do not believe in the importance of social change. Consequently, they are also less intentioned to actively participate in the movement.

It is also noteworthy that benevolent and hostile beliefs about men had positive and negative relations, respectively, with the perception that the #MeToo campaign put men under challenging positions and pitted the sexes against each other. This suggests that women endorsing benevolent views of men might also be more prone to accept the criticisms advanced against #MeToo, according to which the movement impedes "the freedom to annoy, which is essential to sexual freedom" [57] or resembles a "witch-hunt" "where every guy in an office who winks at a woman is suddenly having to call a lawyer to defend himself" [58]. Of course, the same criticisms would probably be rejected by women who feel more antipathy toward men as a group for their dominant position in society and

how they assert control within intimate relationships. However, in both cases, the (higher or lower) perception of the detrimental effects of #MeToo was not related to women's intentions to participate in the campaign. This result matches what we found for men participants. In other words, for both men and women, the effectiveness of the #MeToo campaign in helping the victims of sexual abuse seems more critical than perceiving the campaign as more or less damaging for their decision of whether to join the movement and actively participate in actions aimed at promoting women's rights.

Overall, we should notice that our findings somehow diverge from previous studies, according to which people's negative beliefs about the outcome of the #MeToo campaign could dampen women's opportunities and rights [6,8]. However, they are also perfectly in line with models of collective actions, according to which the primary antecedents of collective actions are perceiving discrimination against the disadvantaged group, feeling that this discrimination is unjust, and perceiving that collective action could be helpful for social change [46]. In this vein, we believe the present studies might have important implications for men's involvement in feminist collective action. Previous research has shown that men are less likely to participate in activities that promote women's rights and autonomy [45] because they are less sensitive to gender inequalities [10], less likely to identify as feminists [58,59], and more likely to believe that gains in women's rights are tied to increased discrimination against men [59]. We further showed that men's participation in the #MeToo campaign, and possibly in other feminist movements, could be increased by promoting educational and political actions to change the shared view that men and women have prescribed societal roles. On women's part, we believe that the present findings complement Radke et al.'s [48] contention that women face barriers to participating in feminist collective action when they endorse sexist beliefs and fail to perceive the efficacy of the movement to achieve the goal of the group. Thus, women who are generally in favor of gender equality or might personally benefit from participation in feminist collective action could refrain from endorsing these actions because of their own internalized sexism and acceptance of the stigmatized representation of feminism.

This has a twofold practical implication. First, achieving the goal of reducing sexual harassment, violence, and abuse against women needs the de-radicalization of sexist ideologies. The shared view that women are more suited to help, nurture, and take care of others and thus to domestic, lower status roles and that, contrarily, men should cover higher status positions because of their higher competence, ambitiousness, and inclination to take risks [60] are detrimental to our society. Second, supporters of the #MeToo movement should strive to disseminate its achievements and the positive changes that the movement brought about to increase people's awareness of its beneficial effects on the victims of sexual abuse and society at large. Among these, we might cite the laws prohibiting the use of nondisclosure agreements in sexual misconduct cases passed in several US states, the legal help provided to thousands of people to seek justice, and the rising of associations aimed at helping the victims of sexual harassment.

When dealing with sexual assault and abuse, one factor often tackled in intervention is women's empowerment. This is undoubtedly useful, especially when the abuse has already taken place and the victim needs support to regain control of their life. Yet, empowerment may be less of a prevention tool. Taken together, the findings of these studies show that ambivalence towards genders is a crucial factor to consider when dealing with the issue of gendered violence. In other words, collective awareness and action may have to start from the deconstruction of gender stereotypes prescribing different roles to different people solely based on their gender.

### 7.2. Limits and Future Directions

The present research has several limitations. First, it relies on cross-sectional and correlational data, so the causal directions of the effects are difficult to discern. However, we should note that previous research has extensively demonstrated the link between ambivalent sexism and attitudes toward sexual harassment. Therefore, it is very likely that

the same causal direction might apply to the relation between ambivalent sexism, attitudes toward the #MeToo movement, and behavioral intentions.

Second, the studies have been conducted in Italy, where, as mentioned, the #MeToo campaign has been extensively criticized, and traditional gender roles are still pervasive. For this reason, it would be interesting to test the replicability and validity of our findings in different cultural contexts characterized by a less pronounced gender gap and a more positive view of feminist movements. Future research could also consider other factors surrounding gender issues, such as men's and women's identification with their group and feminist ideology, their awareness of gender inequality, and previous experience of sexual harassment or assault, on the willingness to support the #MeToo movement.

A third limitation of the present studies concerned the measures we used. As mentioned in the method section, the reliabilities of benevolent sexism toward women and hostile beliefs about men scales are relatively low. Similar limitations have been identified in previous studies and can be attributed to the type of scales (as indicated in the methods section) [52]. Future research should use different instruments not only to overcome the reliability issue but also to seize current evolutions of ambivalence towards men and women. Moreover, we measured intentions to engage in collective action that supports the #MeToo campaign instead of self-reports of past behavior or actual behaviors. However, this choice takes account of practical limitations and opportunities, in line with most of the research on collective participation that generally relies on proxies for actions [46]. Considering that we asked participants about their willingness to participate in actions that involve relatively low costs, we believe that our measures represented a good indicator of the endorsing of collective action that supports the #MeToo movement.

## 8. Conclusions

Harassment toward women is still a very ubiquitous and dramatic reality. As put in the fifth UN Sustainable Development Goal, societies must aim to eliminate all sorts of violence against women and strive for gender equality. In this regard, campaigns such as #MeToo are the foundation for radical change, both by raising awareness of the pervasiveness of sexual harassment and abuse worldwide and helping the victims to speak up and obtain justice. Given the relevance of such movements, it is of the utmost importance that both women and men support them, which can be done by enacting several actions, from sharing information about it on social media and talking about it with other people, to participating in informative public events, protests, and marches. The present research sheds light on why some people are more willing than others to engage in such actions in support of #MeToo. Specifically, it shows that when men and women endorse traditional gender roles and accept men's consequent domination over women, they also tend to think that campaigns such as #MeToo are hardly effective in helping the victims of sexual harassment and achieving social change. As a consequence, they are less willing to join these movements. Overall, the present results seem to uphold the need for a societal shift that eradicates traditional gender roles and the related sexist ideologies that legitimize men's dominant societal stand. Women and men should be encouraged to recognize and confront subtle forms of sexism, including those that try to jeopardize the effectiveness and credibility of feminist movements, which, among other things, draw public attention to inequities and provide political pressure to establish women's rights.

**Supplementary Materials:** The following supporting information can be downloaded at: https: //www.mdpi.com/article/10.3390/su141912294/s1, Supplementary material presents the factorial analyses results of Study 1 and Study 2.

**Author Contributions:** Conceptualization, S.M. (Silvia Moscatelli), S.M. (Silvia Mazzuca), S.C. and M.M.; methodology, S.M. (Silvia Moscatelli), S.M. (Silvia Mazzuca), S.C. and M.M.; formal analysis, S.M. (Silvia Moscatelli), S.M. (Silvia Mazzuca) and S.C.; investigation, M.M., S.M. (Silvia Moscatelli) and S.M. (Silvia Mazzuca); writing—original draft preparation, M.M. writing—review and editing,

S.M. (Silvia Moscatelli); S.C. and S.M. (Silvia Mazzuca). All authors have read and agreed to the published version of the manuscript.

**Funding:** This research received no external funding.

**Institutional Review Board Statement:** The study was conducted in accordance with the Declaration of Helsinki, and approved by the Bioethics Committee of the University of Bologna (Prot. 117001-29 May 2019).

**Informed Consent Statement:** All subjects gave their informed consent for inclusion before they participated in the studies.

**Data Availability Statement:** The data supporting the reported results are available under request to the first author.

**Conflicts of Interest:** The authors declare no conflict of interest.

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
