# Peer review of "Behind the Lines of #MeToo: Exploring Women’s and Men’s Intentions to Join the Movement"

_sustainability, doi:10.3390/su141912294_

Round 1

Reviewer 1 Report

Overall an incredibly necessary and insightful study with a comprehensive literature review. Below are my suggestions/comments: 

1) I'm unsure about the grammatical usage of the phrase "the #MeToo". Stylistically, when reading it, it reads awkwardly. It would read better if it were either just "#MeToo" or "the movement #MeToo" or as you wrote on page 1 line 44 "the #MeToo campaign".

2)  Line 63-64, " women’s ambivalent sexism toward men" and you cited Glick et al 2002 to support your statement. The concept of women's sexism toward men is highly contested, in fact Glick et al 2002’s measure of ambivalent sexism is designed to examine positive and negative beliefs about women:  "Glick and Fiske (1996, 2001) have developed a measure, the Ambivalent Sexism Inventory (ASI), which distinguishes between sexist beliefs that are subjectively positive versus negative toward women (p.1)" In fact, the entire concept of reverse sexism (women’s sexism toward men) has been likened to reverse racism. In order for you to move forward with this research project you must address these issues; either by defining what you mean by women’s sexism toward men and providing a discussion around the controversial nature of that statement showing both sides of the debate and showing evidence for why you choose this particular concept, or, and perhaps the easier (and less controversial) option would be to change the wording from “sexism” to something like “prejudice” or “discrimination” (or even something like “preconceived notions”).

3) Line 128, remove “themselves”.

4) Subheading 3, line 146. Again I must stress the fact that the notion of sexism toward men is highly contested among feminists and sociologists. Florence Rush views reverse sexism as a misogynistic reaction to feminism. Men’s rights activists such as Warren Ferrell and Jordan Peterson are passionate advocates against feminism and they use the theory of reverse sexism to support their views. Do you really want your research to be associated with the men’s rights movement? If that is what you’re arguing, then you need to state that very clearly, and you will need to go into the literature of reverse sexism. But if it’s just a matter of word choice, then just change the wording so the message of your paper does not get muddled.

5) The entire paragraph (from line 191 onwards) needs rephrasing. You cite Glick et al again to support your views on women’s sexism towards men, however the article you cite does not use the word sexism in this context, so your use of this citation is misleading. Their 1999 article is entitled: “The Ambivalence Toward Men Inventory: Differentiating Hostile and Benevolent Beliefs About Men.” I suggest that you borrow their wording and keep away from referring to women’s distrust of men as sexism toward men.

6) Again, you cite another paper, this one by Rollero and Tartagli (2019) to support your notion of women’s sexism toward men, if you read their article carefully you will find that their wording does not support your claims. In fact, they say “results showed that hostile sexism toward women fostered the endorsement of each myth, whereas benevolence toward men enhanced the myth ‘He didn’t mean to’” (my italics). Please be careful when you use citations not only so as to not mislead your readers, but also so as not to misrepresent the academics you’re citing.

7) On the same point, on line 275 you cite Chapleau et al 2007. However, in their paper they give a clear definition and explanation with what they mean by sexism toward men:

“Ambivalent sexism toward men and rape myth acceptance

Another consideration in determining perceptions of blame in sexual assault cases is deference toward the male perpetrator. Ambivalent sexism toward men reflects women’s conflicted relationship with the more powerful out-group (Glick and Fiske 1999). Women resent men for their higher social status, yet heterosexual women typically depend on men as protectors, providers, and romantic partners. To resolve this conflict, hostile sexist attitudes toward men are the “safe jabs” women take at men that allow women to express their dissatisfaction with patriarchy but yet acknowledge the inevitability of male domination. Benevolent sexism toward men reinforces women’s need to seek the company of men, charges women with the responsibility of taking care of men, and justifies men’s higher status as preordained.”

It would be incredibly beneficial to your paper if you (very early on in your paper) define and explain exactly what you mean by sexism toward men. That would place your research far far away from the notion of reverse sexism. I believe that you have conducted some strong research, however in your assumption that everyone is aware of this theory the message of your paper is coming across as a bit confusing.  

8) What about the ethnicity of the women who participated? It's worth mentioning whether they were all white Italian or whether there were ethnic minorities involved. 

9) Line 599, perhaps worth a footnote to mention that Weinstein has now been imprisoned for his crimes against women? 

10) line 626. An example of the models would be great (I know you provided a reference, but some elaboration for some context would be really insightful.

Author Response

Overall an incredibly necessary and insightful study with a comprehensive literature review. Below are my suggestions/comments: 

Response. We would like to thank the Reviewer for the encouraging comments on our research.

  1. I'm unsure about the grammatical usage of the phrase "the #MeToo". Stylistically, when reading it, it reads awkwardly. It would read better if it were either just "#MeToo" or "the movement #MeToo" or as you wrote on page 1 line 44 "the #MeToo campaign".

Response. Thank you very much for noticing this; we changed the wording according to your suggestions throughout the manuscript.  

2. Line 63-64, " women’s ambivalent sexism toward men" and you cited Glick et al 2002 to support your statement. The concept of women's sexism toward men is highly contested, in fact Glick et al 2002’s measure of ambivalent sexism is designed to examine positive and negative beliefs about women:  "Glick and Fiske (1996, 2001) have developed a measure, the Ambivalent Sexism Inventory (ASI), which distinguishes between sexist beliefs that are subjectively positive versus negative toward women (p.1)" In fact, the entire concept of reverse sexism (women’s sexism toward men) has been likened to reverse racism. In order for you to move forward with this research project you must address these issues; either by defining what you mean by women’s sexism toward men and providing a discussion around the controversial nature of that statement showing both sides of the debate and showing evidence for why you choose this particular concept, or, and perhaps the easier (and less controversial) option would be to change the wording from “sexism” to something like “prejudice” or “discrimination” (or even something like “preconceived notions”).

Response.

We thank you for these punctual observations. We decided to follow your suggestion and to use Glick and Fiske’s wording of “ambivalence toward men” and “hostile and benevolent beliefs about men”. We hope that this choice will be useful to disambiguate our intentions.

3) Line 128, remove “themselves”.

Response. We removed “themselves”

4) Subheading 3, line 146. Again I must stress the fact that the notion of sexism toward men is highly contested among feminists and sociologists. Florence Rush views reverse sexism as a misogynistic reaction to feminism. Men’s rights activists such as Warren Ferrell and Jordan Peterson are passionate advocates against feminism and they use the theory of reverse sexism to support their views. Do you really want your research to be associated with the men’s rights movement? If that is what you’re arguing, then you need to state that very clearly, and you will need to go into the literature of reverse sexism. But if it’s just a matter of word choice, then just change the wording so the message of your paper does not get muddled.

Response. Thank you again for these comments, of course our intention was not to support a reverse racism/sexism hypothesis. As explained in point 2, we decided to change the wording of ambivalence toward men to avoid any misinterpretation.

5) The entire paragraph (from line 191 onwards) needs rephrasing. You cite Glick et al again to support your views on women’s sexism towards men, however the article you cite does not use the word sexism in this context, so your use of this citation is misleading. Their 1999 article is entitled: “The Ambivalence Toward Men Inventory: Differentiating Hostile and Benevolent Beliefs About Men.” I suggest that you borrow their wording and keep away from referring to women’s distrust of men as sexism toward men.

Response. We followed your suggestion, as explained at point 2.

6) Again, you cite another paper, this one by Rollero and Tartagli (2019) to support your notion of women’s sexism toward men, if you read their article carefully you will find that their wording does not support your claims. In fact, they say “results showed that hostile sexism toward women fostered the endorsement of each myth, whereas benevolence toward men enhanced the myth ‘He didn’t mean to’” (my italics). Please be careful when you use citations not only so as to not mislead your readers, but also so as not to misrepresent the academics you’re citing.

Response. Thank you for noticing this. We think that the change in our wording fits with the content of this paper.

7) On the same point, on line 275 you cite Chapleau et al 2007. However, in their paper they give a clear definition and explanation with what they mean by sexism toward men: “Ambivalent sexism toward men and rape myth acceptance Another consideration in determining perceptions of blame in sexual assault cases is deference toward the male perpetrator. Ambivalent sexism toward men reflects women’s conflicted relationship with the more powerful out-group (Glick and Fiske 1999). Women resent men for their higher social status, yet heterosexual women typically depend on men as protectors, providers, and romantic partners. To resolve this conflict, hostile sexist attitudes toward men are the “safe jabs” women take at men that allow women to express their dissatisfaction with patriarchy but yet acknowledge the inevitability of male domination. Benevolent sexism toward men reinforces women’s need to seek the company of men, charges women with the responsibility of taking care of men, and justifies men’s higher status as preordained.”

It would be incredibly beneficial to your paper if you (very early on in your paper) define and explain exactly what you mean by sexism toward men. That would place your research far far away from the notion of reverse sexism. I believe that you have conducted some strong research, however in your assumption that everyone is aware of this theory the message of your paper is coming across as a bit confusing.  

Response

We followed your suggestion and explained in the very first introduction what we meant for women’s ambivalence toward men: “Study 2 examined the correlates of intentions to support the #MeToo campaign among women. Specifically, it focused on women’s ambivalence towards men, which is expressed through hostile resentment toward male dominance on the one hand and respectful benevolence toward men who serve in the roles of providers and protectors on the other (Glick e Fiske 1999)” lines 64-69.

We also deepened this explanation in lines 107-119: “Besides sexist ideologies regarding women, attitudes towards sexual harassment could be affected by ambivalent beliefs about men. According to Glick and Fiske [12], women respond to male power and dominance by holding both hostile and benevolent beliefs about men. Hostility conveys resentment toward men’s higher status and sexual aggressiveness. In contrast, benevolent beliefs about men represent admiration for their role as protectors and providers by expressing the maternalistic view that men need women’s care [24]. It has been noted, however, that even ambivalence toward men can be a means through which men’s domination over women within a society is justified and maintained: Indeed, hostile beliefs about men might represent a “safe jab” that al-lows expressing discontent on how men treat women while acknowledging their inevitable higher status. At the same time, the benevolent portrayal of men as needing maternalistic care implicitly recognizes men’s superiority in less private (and more socially acknowledged) domains [12,14].”

8) What about the ethnicity of the women who participated? It's worth mentioning whether they were all white Italian or whether there were ethnic minorities involved. 

Response. Participants were all white Italian. We added this information.

9) Line 599, perhaps worth a footnote to mention that Weinstein has now been imprisoned for his crimes against women? 

Response. We added this information in the introduction where we describe the movement (lines 141-143): “In 2020 Weinstein was sentenced to 23 years in prison for rape and sexual assault, and he is currently facing a second trial in another State.”

10) line 626. An example of the models would be great (I know you provided a reference, but some elaboration for some context would be really insightful.

Response. We added a brief description of Van Zomeren et al.’s model as follows “However, they are also perfectly in line with models of collective actions that according to which the primary antecedents of collective actions are perceiving discrimination against the disadvantaged group, feeling that this discrimination is unjust and perceiving that collective action could be useful for social change.” Lines 651-655.

Reviewer 2 Report

First, I would like to thank you for the opportunity to review this paper, which aims to identify possible motivations that bring men and women closer to the #MeToo movement, through two different studies. The paper is based on good assumptions but needs some editing before it can be published.

Title/Abstract/Keywords: I would suggest not using terms that can be also found in the title as keywords since they are already indexed in search engines.

Introduction:

The introduction is very long and redundant, difficult to follow despite being divided into paragraphs. In particular, the differences between the two studies should be included in the aims. Paragraph 2 on the #MeToo movement is verbose and distracts from the focus of the research, as does paragraph 3, which could be integrated with the previous one. I would restructure the introduction like this:

- Gender violence and its causes (including ambivalent sexism).

- The #MeToo and recent feminist movements

- Research aims.

Studies

It would be preferable for the studies to be numbered 1 and 2 and not 1a and 1b; each study should present the specific research objectives, which should be moved from the initial "Overview" block.

Procedures:

What items are used for the "questions concerning the #MeToo movement"? They should also be included for clarity as supplementary material, specifying to which of the variables each item corresponds, also explaining how the three variables were created (were factorial analyses done? How do we know if the items measure an assimilated construct? Without this information, we cannot interpret subsequent analyses). A Data Analysis block is missing (the information is now in Results). Has any preliminary analysis of the data been done? Are sample normality conditions met? This is important information to be able to interpret subsequent data. It is not necessary to include the results of these analyses in the text, but just include a sentence that allows the reader that the pretreatment was done.

Minor comments:

Check consistency in uppercase and lowercase (violence against women is sometimes uppercase, sometimes lowercase) and some typos and spelling in English.

Author Response

Title/Abstract/Keywords: I would suggest not using terms that can be also found in the title as keywords since they are already indexed in search engines.

Response. Thank you very much for this suggestion.

Introduction:

The introduction is very long and redundant, difficult to follow despite being divided into paragraphs. In particular, the differences between the two studies should be included in the aims. Paragraph 2 on the #MeToo movement is verbose and distracts from the focus of the research, as does paragraph 3, which could be integrated with the previous one. I would restructure the introduction like this:

- Gender violence and its causes (including ambivalent sexism).

- The #MeToo and recent feminist movements

- Research aims.

Response. Thank you for your suggestions on how to reframe the Introduction. We shortened it (it is now 2399 words), and we also changed the order of the topics following your indication. We decided to rename the headings as follows:

  • Ambivalence toward Women and Men as an Antecedent of Attitudes toward Sexual Harassment (we preferred this title because we do not review the general literature on gender violence but ambivalent sexism as antecedent of attitudes toward sexual harassment)
  • The movement #MeToo (we preferred this title because we just briefly cited other feminist movements, but we did not discuss them).
  • Research aims (instead of Overview)

Studies

It would be preferable for the studies to be numbered 1 and 2 and not 1a and 1b; each study should present the specific research objectives, which should be moved from the initial "Overview" block.

Response. We followed your indications and changed the numbers of the studies as well as moved the specific objectives and hypotheses of the two studies from the Research Aims section to each study section.

Procedures:

What items are used for the "questions concerning the #MeToo movement"? They should also be included for clarity as supplementary material, specifying to which of the variables each item corresponds, also explaining how the three variables were created (were factorial analyses done? How do we know if the items measure an assimilated construct? Without this information, we cannot interpret subsequent analyses). A Data Analysis block is missing (the information is now in Results). Has any preliminary analysis of the data been done? Are sample normality conditions met? This is important information to be able to interpret subsequent data. It is not necessary to include the results of these analyses in the text, but just include a sentence that allows the reader that the pretreatment was done.

Response.

Thank you for these suggestions on how to make the Methos and Results sections clearer.

The items we used for all the questions concerning the #MeToo movement were already spelled in the previous version of the manuscript and concern the perception of its beneficial and detrimental effects and the intentions to participate in the movement. However, we also added the entire spelling of the items in a Supplementary Material file with the results of the factorial analyses conducted to determine the factor structure of measures concerning the #MeToo campaign.

We added Data Analysis sections in both studies to explain which analyses were performed: preliminary analyses (with the specification that the normality assumptions were not violated), factor analyses, correlations, descriptive analyses, and SEMs.

Minor comments:

Check consistency in uppercase and lowercase (violence against women is sometimes uppercase, sometimes lowercase) and some typos and spelling in English.

Response. We extensively checked for typos and inconsistencies.

Reviewer 3 Report

Fighting Sexual Harassment and Violence: The Antecedents of Women's and Men's Intentions to Join the #MeToo

This is an interesting contribution to the journal, but I have some concerns regarding its contribution to scholarship, and I believe some changes would improve the quality of the article thus increasing its chances of being published:

1.     Title: Fighting Sexual Harassment and violence – I don’t believe this reflects what was measured in the studies, I would suggest author to reformulate the title. Determinants perhaps? 

2.     Abstract: please rewrite abstract according to the journal’s format, informing your readers about the methodologies involved.

3.     Authors must rewrite the introduction, as it lacks parsimony. Almost 4000 words it should be more synthetized. Also, a brief note on the etiology of sexism in Italy would be relevant.

4.     Lines 331 and 447. Is this a Cronbach’s alpha? =.53? If so, this value is unacceptable, and the subscale should not be used.

5.     Please include an implications section, focusing on mental health policies and the prevention of sexual violence.

Best wishes.

Author Response

  1. Title: Fighting Sexual Harassment and violence – I don’t believe this reflects what was measured in the studies, I would suggest author to reformulate the title. Determinants perhaps? 

Response.

Thank you for noticing this. We agree that the title needed reformulation. To avoid inconsistencies between the title and what we did in the studies we proposed this new title: “Behind the lines of #MeToo: Exploring Women's and Men's Intentions to Join the Movement”

  1. Abstract: please rewrite abstract according to the journal’s format, informing your readers about the methodologies involved.

Response.

We added information about the method in the abstract. We specified that the studies have a correlational nature and wrote: “Men (Study 1) and women (Study 2) participants were asked to answer questions concerning their ambivalent beliefs about women and men respectively, their perception of the beneficial and detrimental effects of #MeToo and their intentions to participate in the campaign”.

  1. Authors must rewrite the introduction, as it lacks parsimony. Almost 4000 words it should be more synthetized. Also, a brief note on the etiology of sexism in Italy would be relevant.

Response.

We shortened the introduction, which now counts 2399 words. We also changed the order of the topics discussed following Reviewer 2’s suggestions and merged different parts. Following Reviewer 1’s indications, we elaborate more on women’s benevolent beliefs about men. Finally, we added information about sexism in Italy in the overview section. We were not completely sure on what you meant by “etiology” in this case, but we reported recent data on how much gender stereotypes are spread in Italy and on Italians’ attitudes toward sexual harassment. We also reported the results of recent studies that examined how sexist views are shared in Italian media. Finally, we made reference to the culture of honor, which could be a further explanation of why sexism in Italy is still widespread. We hope that this further information could satisfy your request.

This is what we wrote: “Recent national reports [36] indicated that the most widespread gender stereotypes among Italians are: “Being successful at work is more important for men than for women,” “Men are less suited for housework,” and that these stereotypes are shared by the 58.8% of the population. For what concerns sexual violence, 39.3% of Italians believe that a woman could avoid sexual intercourse if she really wanted to, 23.9% declared that women could provoke sexual violence by their clothing, and 15.1% think that a woman who suffered sexual abuse when she was drunk or on drugs is at least in part responsible. These stereotypes are evident in Italian media, whereby women are more often portrayed as objectified, attractive, and dressed in a sexualized way than in less masculine and gender-unequal countries [35]. Unfortunately, these stereotypic representations have worsened since the nineties [37]. It is also important to consider that Italy still represents, to a certain extent, a “culture of honor” that valorizes norms of superiority and toughness for men and norms of modesty and shame for women [38,39]. According to these norms, women should avoid behaviors that might threaten a family’s reputation, and men’s violence against women might be even justified when aimed at preserving the integrity of the man and the family [39,40]. Lines 200-215.

Is this a Cronbach’s alpha? =.53? If so, this value is unacceptable, and the subscale should not be used.

We agree that Cronbach’s alphas for the measures of benevolent sexism toward women (Study 1) and hostile beliefs about men (Study 2) were relatively low. Such low values can be at least partially due to the use of ultra-short (3 items) versions of the scales. Indeed, Bosson et al., 2021, in their paper in the European Journal of Social Psychology, used the same short versions of the scales reporting the very same problem with internal reliability values. Moreover, as Hinton et al. (2004) reported, with scales of few items, Alpha of .50 can be considered acceptable (see also Pallant, 2011).

We recognize that our results related to the two measures above should be interpreted cautiously and we add this caution note in the Limits section. However, we think that including benevolent sexism toward women and hostile beliefs about men in the two studies is still relevant to the overall aims of the research and provides interesting results.

Thus, we maintained such results in the revised version of the manuscript while justifying this choice in the Method section and mentioning these weaknesses in the discussion (“A third limitation of the present studies concerned the measures we used. As mentioned in the method section, the reliabilities of benevolent sexism toward women and hostile beliefs about men scales are relatively low. Similar limitations have been identified in previous studies and can be attributed to the type of scales (as indicated in the methods section) [52]. Future research should use different instruments not only to overcome the reliability issue but also to seize current evolutions of ambivalence to-wards men and women” – lines 707-712).

Nevertheless, following your suggestion, we re-run the models of the two studies eliminating the two measures. The fit indexes are good for both Study 1, CFI = .942, TLI = .924, RMSEA = .078, CI [.066, .090], and Study 2, CFI = .980, TLI = .973, RMSEA = .052, CI [.035, .067].

As in the manuscript model, in this new model of Study 1, hostile sexism has significant direct relation with the measures of perceived beneficial effects, perceived detrimental effects, and intention to participate in the #MeToo campaign. It also has a significant indirect link with the intention to participate variable through perceived beneficial effects. Concerning Study 2, the measure of benevolent beliefs toward men has a significant negative direct effect on the perceived beneficial effects of the #MeToo campaign and a positive direct effect on the measure of perceived detrimental effects, whereas the direct link with intention to participate is not significant. The benevolent beliefs measure also has an indirect effect on the intention to participate in #MeToo through the perception of its beneficial effects, as in the original model.

Although we think that maintaining the original models in both studies renders the research theoretically completer and more interesting for readers – despite the low but acceptable values of the two alphas – we are willing to modify the studies reporting the reduced models described above if the Editor thinks this is the best option.

References

Bosson, J. K., et al. (2021). (2021). Precarious manhood beliefs in 62 nations. Journal of Cross-Cultural Psychology, 52(3), 231–258. https://doi.org/10.1177/0022022121997997

Hinton, P., McMurray, I., & Brownlow, C. (2014). SPSS Explained (2nd ed.). Routledge. https://doi.org/10.4324/9781315797298

Pallant, J. (2011). SPSS survival manual: A step by step guide to data analysis using SPSS (4th ed.). Maidenhead, Australia: Open University Press/McGraw-Hill.

  1. Please include an implications section, focusing on mental health policies and the prevention of sexual violence.

Response.

We tried to expand the practical implications, even if we were not able to focus on specific mental health policies for lack of expertise on this domain. We added the following implication:  When dealing with sexual assault and abuse, one factor that is often tackled in intervention is women's empowerment. This is undoubtedly useful, especially when the abuse has already taken place and the victim needs support in taking control over their life back. Yet, empowerment may be less of a prevention tool. Taken together, the findings of these studies show that ambivalence towards genders is a key factor to consider when dealing with the issue of gendered violence. In other words, collective aware-ness and action may have to start from the deconstruction of gender stereotypes pre-scribing different roles to different people solely based on their gender.” Lines 683-690.

We hope that this comment could be relevant to what you asked for.

Round 2

Reviewer 3 Report

Thank you for implementing all the requested changes to the manuscript.

I believe the article is now fit for publication.

Best wishes.